# At the Peak of the Second Wave of COVID-19, Did Millennials Show Different Emotional Responses from Older Adults?

**DOI:** 10.3390/ijerph19105908

**Published:** 2022-05-12

**Authors:** Aurélie Wagener, Céline Stassart, Anne-Marie Etienne

**Affiliations:** Research Unit for a Life-Course Perspective on Health and Education (RUCHE), Health Psychology, Department of Psychology, Faculty of Psychology and Science of Education, University de Liege, Rue de l’Aunaie, 30, 4000 Liege, Belgium; cstassart@uliege.be (C.S.); am.etienne@uliege.be (A.-M.E.)

**Keywords:** COVID-19, Millennials, emotions, emotion regulation strategies, intolerance of uncertainty, environmental satisfaction

## Abstract

The COVID-19 pandemic disturbs mental health. Indeed, higher levels of negative emotions and anxiety, along with lower levels of positive emotions and well-being, have been highlighted. As a result, individuals need to regulate these psychological states in a context of uncertainty and daily restrictions (e.g., access to leisure activities, social contacts) or modifications of livelihood (e.g., working modalities). Overall, psychological reactions to the pandemic have been shown to differ based on one’s age. The purpose of this research is to compare psychological reactions to the pandemic between Millennials (born between 1981 and 1996) and Older Adults. The generation’s prism seems relevant as members of specific generations are shaped (e.g., views of the world, the future) by their similar experiences. Ninety-four Millennials and 129 Older Adults, recruited in the general population, participated to an online survey assessing emotions, emotion regulation strategies, environmental satisfaction, and intolerance of uncertainty. Results show that Millennials experience higher levels of negative emotions along with higher levels of worry and rumination than Older Adults. Millennials also report a higher level of joy. Nonetheless, more Older Adults engage themselves in novel activities. Overall, findings confirm previous ones and indicate the need to offer effective clinicals tool to prevent mental health worsening.

## 1. Introduction

### 1.1. Health Context

Since the beginning of 2020, the World faces the COVID-19 pandemic which is characterized by “waves” of emergence of the virus and its variants (e.g., Delta or Omicron). The first wave occurred during spring 2020, the second during autumn 2020, and others of varied intensity in 2021.

As a major worldwide health threat, the COVID-19 pandemic has incontestably disrupted our everyday lives. More concretely, working and schooling modalities changed to homeworking and homeschooling. Social contacts (e.g., dinner with friends) as well as access to leisure activities or hobbies (e.g., cinema or sports) were strictly limited. Overall, these restrictions aimed at restraining the spread of the virus to protect health, at a societal level. Nonetheless, they led to consequences in terms of mental health issues.

### 1.2. Psychological Consequences

Since the onset of the pandemic, an important number of studies have been interested in its psychological impacts and showed increases in rates of mental health disturbances. Indeed, in the general population, the prevalence of psychological distress increased from 18.3% to 28.3% between 2018 and April 2020 [1,2]. In line with the previous, Blix et al. (2021) [3] and Taylor et al. (2020) [4] evidenced that more than 25% of the general population experienced psychological distress, meaning that their psychological symptoms (e.g., anxiety) exceeded clinical significance. Beyond an increased prevalence of psychological distress, mental health has been characterized by decreases in positive emotions and quality of life [5,6,7,8] as well as increases in negative emotions, anxiety, fear, stress, depression, and sleep disturbances [8,9,10,11,12,13,14].

The aforementioned negative consequences of the COVID-19 pandemic could be partly explained by the uncertainty inherent to the health situation (e.g., course of the pandemic and length of the disease) [15,16]. Indeed, facing situations inducing uncertainty, some individuals might experience “uncertainty distress” (i.e., “*the subjective negative emotions experienced in response to the as yet unknown aspects of a given situation*”) [17]. Situations of uncertainty and their related distress might, in turn, render these individuals intolerant to uncertainty [17]. In the specific context of the pandemic, different authors evidenced that intolerance to uncertainty is a significant predictor of anxiety and depression [15,16] and, more generally, of negative emotions [8]. Therefore, research investigating psychological reactions to the pandemic should thoroughly address this variable.

When facing emotional distress, individuals naturally tend to alleviate them by applying emotion regulation strategies [18]. Adaptive emotion regulation strategies (e.g., acceptance and positive reappraisal) have been linked to higher levels of well-being and more satisfactory quality of life, while non-adaptive ones (e.g., worry and self-blame) have been related to lower levels of quality of life and well-being [19,20]. In the specific context of the pandemic, Wagener, Stassart, and Etienne (2022) [8] reported an increase in the recurse to “acceptance”. Based on its definition as “*thoughts of accepting what you have experienced and resigning yourself to what has happened*” [20], this result indicates a healthy response to the situation while accepting the situation should be distinguished from the emotional reactions to the situation [8].

Beyond emotions, assessing environmental satisfaction seems relevant in the context of the pandemic. Environmental satisfaction is defined as “*one’s perception of the positive or negative value of environmental experiences and activities available in its environment*” [21]. Then, life domains (e.g., family and social relationships, professional or education area, and leisure) can be characterized by different levels of environmental satisfaction. In a sample of adults from the general population, a decrease in environmental satisfaction since the pandemic was shown [8]. The authors hypothesized that this phenomenon might be explained by lockdown’s measures which caused the majority of establishments to close (e.g., non-essential shops such as bookshops or garden centers, cinemas, and gyms), and that almost all activities ceased (e.g., leisure and artistic activities) along with the strict limitation of social contacts. Further, they showed that environmental satisfaction was the most predictive variable of both positive and negative emotions during the first wave of COVID-19 [8]. These results are coherent with the literature on behavioral models of depression according to which lower levels of engagement in activities—resulting in lower levels of environmental satisfaction—are positively related to symptoms of depression [22,23,24,25] while higher levels of engagement in activities are related to well-being [26,27]. As these mechanisms are well-established, research investigating psychological reactions to the pandemic should thoroughly address their relevance in this specific context.

While it is generally accepted that the COVID-19 pandemic challenges everyone, regardless of age, this latest variable appeared to influence the experience with the pandemic (e.g., [28,29]). Indeed, several studies showed that young adults were the most at risk of psychological distress in different countries such as India, Spain, Portugal, Italy, Iran, and Belgium (e.g., [13,16,30,31]). More precisely, Huang and Zhao (2020) [10] highlighted that adults under 35 years experienced higher levels of anxiety and depression than older adults. Klaiber et al. (2021) [32] also reported that, overall, negative effects were higher in young adults and that positive ones were lower than in older adults.

Following the previous, investigating how generations cope with the COVID-19 pandemic seems of interest. Generations can be defined as cohorts of individuals born in the same space–time, and consequently, go through similar steps of their life simultaneously. Further, as explained by the Pew Research Center [33], it is assumed that every generation faces specific challenges and issues such as world events or technological, economic, and social shifts. These experiences shape one’s world views through interactions with the life cycle and aging process. As the COVID-19 pandemic is challenging for everyone, it seems relevant to investigate psychological reactions through the generation’s prism. Indeed, it is likely that generations perceive the pandemic differently. Even though the effect of age on the pandemic’s experience has been widely investigated, very few studies adopted a generational perspective. We argue that the generation prism is relevant as it is highly possible that the experience of the pandemic and its related needs are different from one generation to another. This original perspective might lead to novel knowledge regarding the pandemic. Concretely, the current study focuses on Millennials—born between 1981 and 1996—to compare them to Older Adults belonging to Generation X and Baby Boomers, born before 1981. Millennials were aged between 24 and 39 years at the time of the present study (i.e., November 2020). In other words, they were mostly considered as young adults who were likely facing challenges in their personal and interpersonal development, education, and professional career. Millennials’ growth has been marked by events such as the 9/11 terrorist attacks, wars in Iraq and Afghanistan, terrorist attacks in Europe, and migratory crises. More generally, they have grown up in a world where climate changes accelerate (e.g., melting of the ice caps, floods, and forest fires). They also grew up in an era of the rapid evolution of technology (e.g., mobile devices and social media) and faced, more specifically, the explosion of the Internet. Overall, this context rendered these “digital natives” lifelong users of the Internet and media and very active in the development and use of novel technologies.

### 1.3. Aims and Hypotheses

We live in an era of health, ecological, economic, or social crises which are accompanied by psychological consequences that can weaken the population’s mental health. Therefore, it seems quite worthy to understand how individuals react to those crises to help them in the most appropriate and effective way. From a public health perspective (e.g., an outbreak of an infectious disease such as COVID-19), understanding psychological reactions is highly relevant as, as mentioned by Cullen, Gulati, and Kelly (2020) [9], those reactions are critical in “*shaping both spread of the disease and the occurrence of emotional distress during and after the outbreak*”.

As the scientific literature evidenced that adults under 35 years were more burdened than older ones in their experience of the pandemic [13], assessing Millennials’ reaction, in particular, seems relevant. Further, focusing on the second wave of the COVID-19 pandemic which occurred during the autumn 2020, the current study aims to contribute to the still growing literature that seeks to better understand psychological consequences of the COVID-19 pandemic. Additionally, our focus on the second wave of COVID-19 meets the suggestions of Salmon et al. (2022) [34], according to whom the effects of the pandemic on mental health during the second peak are not clear. While our results will concern a specific time, they might be extended to times of crises in general, which appears quite important as other epidemics have already marked history (e.g., Spanish influenza), and because it is quite likely that the World will again be confronted to such sanitary threats.

This research concretely compares Millennials and Older Adults’ mental health during the COVID-19 pandemic by assessing their emotions, emotion regulation strategies, intolerance of uncertainty, and environmental satisfaction through an online survey in the general population. These comparisons will offer an overview of psychological issues depending on the generation’s belonging. From a clinical perspective, the results of the current study might strengthen the need of having at our disposal efficient clinical tools to avoid mental health worsening (e.g., prevention, detection, assessment, and intervention). Further, as reminded by Glowacz and Schmits (2020) [16], groups that are different by their age should be comprehended differently. If our results underline generations’ differences, this will shed light on the need to tailor some—or all—of our clinical interventions depending on age.

We hypothesized that we would replicate previous results. More precisely, a more important psychological distress—in terms of negative effects and intolerance of uncertainty—in Millennials is expected. Our study remains exploratory concerning the other variables (i.e., emotion regulation strategies and environmental satisfaction). In fact, to our knowledge, the influence of generation on emotion regulation strategies and environmental satisfaction in the context of the pandemic has not been studied to date.

## 2. Materials and Methods

The current study replicates the methods used in Wagener, Stassart, and Etienne (2022) [16]. Consequently, the description of the materials and methods is mostly duplicated.

### 2.1. Study Design

Power analyses were performed a priori using G*Power 3.1. [35]. Applied to our t test for independent samples (t tests; means: difference between two independent means—two groups), the power calculation (power of 0.95, α-error of 0.05, an estimated effect size of 0.5) indicated that a sample of 176 participants was requested to detect effects. Applied to our χ2 tests of association (goodness-of-fit tests: contingency tables), the power calculation (power of 0.95, α-error of 0.05, an estimated effect size of 0.5, degree of freedom equal to 1) indicated that a sample of 52 participants was requested to detect effects.

### 2.2. Participants and Procedure

The current sample comprised 213 adults (164 women) from the general population with an average age of 43.88 (range = 24–79, SD = 13.4). From November 2020 to December 2020, recruitment occurred through advertisements on social networks and university websites’ announcements. The study’s aim was described, and a link to an online survey was provided. The survey was hosted by the online study software developed by the Faculty of Psychology, Speech and Language Therapy, and Education of the University of Liege. The survey was composed of the following evaluations: socio-demographic information, activities’ information, emotions, emotion regulation strategies, intolerance of uncertainty, and environmental satisfaction. The administration of these scales was part of a more comprehensive evaluation process that included other self-reported measures such as applying barrier gestures. As the current paper focuses on emotional aspects, the other self-reported measures were not involved, as they rather address health beliefs and behaviors than emotions. The local ethics committee of our college of psychology approved the protocol (approval number: 1920-97), and all participants provided online informed consent.

### 2.3. Measures

#### 2.3.1. Demographic Information

Participants had to report their gender, year of birth, and relationship status, and indicate if they have children. Regarding work situation, they had to specify if their working conditions were identical to those before the COVID-19 pandemic or have been modified (i.e., homeworking or technical lay-off).

#### 2.3.2. Activities Information

Participants were asked if they started novel activities in different life areas, namely: artistic activities (e.g., painting or sculpting); sporting activities (e.g., going cycling or for a walk); gardening; cooking; well-being activities (e.g., yoga or meditation), or COVID-related helping activities (e.g., grocery shopping for someone else or stitching facial masks for other people). These life domains were selected based on the Brief Behavioral Activation Treatment for Depression—Revised version [36]. In this treatment protocol, the authors mention ten life areas which cover a vast majority of activities: family relationships, social relationships, romantic relationships, education/training, employment/career, hobbies/recreation, volunteer work/charity/political activities, physical and psychological health issues, spirituality, and daily responsibilities. Due to lockdown and sanitary restrictions, novel activities in some of these life areas were forbidden (e.g., social relationships and romantic relationships). Therefore, we only assessed the remaining life areas in which individuals were able to start novel activities (i.e., physical and psychological health issues, hobbies/recreation, volunteer work/charity/political activities, and daily responsibilities).

#### 2.3.3. Basic Emotions Scale

The Basic Emotions Scale assesses five clusters of basic emotions: joy, anger, anxiety, sadness, and disgust. Emotions are evaluated through 20 different emotional terms (e.g., happy, frustrated, anxious, depressed, or blameworthy) rated on a 7-point Likert scale (1 = Not at all, 7 = All the time) [37,38]. A score can be calculated for each cluster of emotions by averaging scores of its component emotions. Internal consistencies were satisfactory, as Cronbach alphas ranged from 0.79 to 0.84 [38].

#### 2.3.4. Emotion Regulation Strategies, Intolerance of Uncertainty, and Environmental Satisfaction

These three variables were assessed with items selected from four different self-assessment questionnaires to limit the length of the protocol. The selection of the items was based on their factor loadings on the reference scale: on each scale, the two items with the higher factor loadings were selected. Participants were asked to assess all items on a 5-point Likert Scale (1 = Totally disagree, 5 = Totally agree). A mean was calculated for each variable.

Four emotion regulation strategies (i.e., acceptance, positive reappraisal, dramatization, and rumination) were assessed through items retrieved from the Cognitive Emotion Regulation Questionnaire (e.g., “*I think that I have to accept the situation*”, “*I think I can learn something from the situation*”, and “*I dwell upon the feelings the situation has evoked in me*”) [39,40]. The fifth emotion regulation strategy, namely worry, was assessed through items retrieved from the Penn State Worry Questionnaire (e.g., “*I know I should not worry about things, but I just cannot help it*”) [41]. Factor loadings ranged between 0.77 and 0.79.

Intolerance of uncertainty was assessed through items retrieved from the Intolerance of Uncertainty Scale (e.g., “*I should be able to organize everything in advance*”), with factor loadings ranging from 0.58 to 0.74 [42,43].

Environmental satisfaction was assessed through items retrieved from the Environmental Reward Observation Scale (e.g., “*My life is boring*” or “*It is easy for me to find enjoyment in my life*”), with factor loadings ranging from 0.79 to 0.83 [21,44].

### 2.4. Statistical Analyses

A first comparison of Millennials and Older Adults’ mental health was performed through Student’s t tests for independent samples. A second comparison of Millennials and Older Alduts’ engagement in novel activities was performed through χ2 tests of association.

Alpha was set at 0.05. Nonetheless, as several statistical analyses were performed, adjusted *p* values were calculated to balance the amount of type 1 and type 2 errors. The false discovery rate method for multiple testing used was the Benjamini–Hochtberg indices, which has been shown to be much more powerful than methods that control the familywise error rate (e.g., Bonferroni) [45,46]. Briefly, the false discovery rate controls the expected proportion of falsely rejected null hypotheses. All analyses were performed with Jamovi 1.6.23.0.

## 3. Results

### 3.1. Demographic Data

Table 1 displays the demographic characteristics of the whole sample. Percentages should be understood by columns. Concerning the age, it is noteworthy that the two groups were statistically different (Table 2).

### 3.2. Comparison of Millennials and Older Adults’ Mental Health

Table 2 presents results of the t tests for independent samples, performed to compare Millennials and Older Adults’ mental health (Benjamini–Hochtberg indice = 0.02).

Significant differences were observed between Millennials and Older Adults regarding emotions. More precisely, the level of joy was higher in Millennials. Levels of fear, guilt, and worry were also higher in Millennials. No significant differences were shown for anger and disgust.

Significant differences were observed between Millennials and Older Adults on emotion regulation strategies. Indeed, levels of worry and rumination were higher in Millennials. No significant differences were shown on the other three emotion regulation strategies.

Lastly, no significant differences were evidenced on intolerance of uncertainty and environmental satisfaction.

Overall, effect sizes for the significant differences were in the low to medium range.

### 3.3. Comparison of Millennials and Older Adults’ Engagement in Activities

In line with the assessment of environment satisfaction, the extent to which participants engaged themselves in novel activities since the second wave of COVID-19 was investigated (Table 3). To compare Millennials to Older Adults, χ2 tests of association were performed (Benjamini–Hochtberg indice = 0.02).

Only one relationship between “Activities” and “Generations” was significant. Older Adults were more likely to engage in “COVID-related helping activities” (χ2 (1, *n* = 213) = 10.0, *p* = 0.002).

## 4. Discussion

Psychological reactions are critical in “*shaping both spread of the disease and the occurrence of emotional distress during and after the outbreak*” [9]. Our study is in line with this statement, as it focuses on the mental health of adults from the general population during the second wave of COVID-19 to identify their psychological difficulties which, in turn, might permit clinical recommendations in terms of prevention, detection, assessment, and intervention. As the influence of age on psychological reactions to the COVID-19 pandemic has been demonstrated (e.g., [16,34]), our study offers a comparison of generations. More precisely, Millennials, born between 1981 and 1996, are compared to Older Adults, born before 1981. The rationale underlying this comparison resides in the fact that members of the same generation face similar experiences, events, or shifts which shape their perception of the World, others, and their future. Therefore, it seems likely that Millennials and Older Adults go through the COVID-19 pandemic quite differently.

The present study firstly aimed at comparing Millennials and Older Adults’ mental health during the COVID-19 pandemic by assessing their emotions, emotion regulation strategies, intolerance of uncertainty, and environmental satisfaction. Regarding emotions, our study replicated previous results, pointing out that the current health situation seems better handled by Older Adults who usually show greater abilities to compensate for stressful events and rely on more developed resilience (e.g., [3,10,13]). Indeed, Millennials reported higher levels of negative effects (i.e., fear and guilt). While feeling negative emotions when facing such a pandemic could be considered as healthy [47], different leads are suggested to explain these results. Firstly, in comparison to Older Adults, younger ones seem to suffer more from loneliness, which has been related to poorer mental health [48,49]. Pieh et al. (2020) [13] suggested that this increased loneliness might be because younger adults face more significant restrictions in their daily routines than older ones (e.g., their own activities such as sports, social activities, or their children’s activities). Secondly, younger adults might go through more uncertainty concerning their working conditions, professional future, and then, financial incomes, than Older Adults who are usually in stable professional situations. Thirdly, Millennials were under social pressure as they were considered as the core group that will importantly contribute to the cessation of the virus’ spreading [50]. Fourthly, these differences might also be explained by the fact that Older Adults improve their ability to solve problems, including emotional ones [51,52]. Therefore, it might be expected that they experienced lower levels of negative effects as they handle such affects more efficiently. Finally, these results might be explained by the fact that Millennials tended to recourse to negative repetitive thoughts (i.e., worry, rumination) more than Older Adults, as shown by our results on emotion regulation strategies. Indeed, mental health has been shown to be harmed by too much time spent thinking about the outbreak [3,10]. This relationship between mood and negative repetitive thoughts is usual and is highly supported by the scientific literature, outside of any pandemic [53].

Interestingly, Millennials also reported higher levels of joy than Older Adults. Several explanations can be discussed regarding this result. The first one finds its source in the two continua model of mental illness developed by Westerhof and Keyes (2010) [54]. According to this model, mental illness and mental health are related but also distinct dimensions. Consequently, Millennials might have experienced both negative and positive emotions in the same period. For instance, under the pandemic’s circumstances, we could imagine that a 34-year-old man could worry about his parents’ health—considered to be at risk of developing a severe form of COVID-19—while in the same day, he will really enjoy having lunch with his kids which was not usual before the pandemic. Further, positive outcomes of the pandemic are more and more highlighted. Indeed, individuals also seem more prone to reassess their priorities and values to live a meaningful life [55]. As a positive relationship exists between pursuing personal values and well-being, it is hypothesized that the feeling of joy might be explained by this phenomenon [56]. Finally, even though Older Adults can improve their ability to solve emotional problems, this result might also evoke the fact that a deterioration of emotional function can be observed in Older Adults [57].

Concerning results on emotion regulation strategies, Millennials reported more negative repetitive thoughts (i.e., worry and rumination) than Older Adults which has already been highlighted outside any pandemic context [58,59]. Nonetheless, it remains relevant to thoroughly address this result. It seems relevant to bring into the discussion the notion of “mental load” which is defined as “*the combination of the cognitive labor of family life*—*the thinking, planning, scheduling and organizing of family members*—*and the emotional labor associated with this work, including the feelings of caring and being responsible for family members but also the emotional impact of this work*” [60]. Based on the previous definition, it seems reasonable to hypothesize that the mental load of parents—and more specifically parents with young children, as Millennials are likely to be—increased during the pandemic. Indeed, these parents—for the most part working full-time job—had to adjust to the lack of boundaries between their professional and personal lives. As explained by Dean et al. (2022) [60], “*homes became more than homes*” as professional activities, schooling activities, and personal activities all occurred in the same walls. Parents were expected to fulfill their professional duties while ensuring that their children were maintaining their investment in their education. This seems all the more unrealistic with young children, which probably describes best Millennials’ life situations. As mental load is mostly cognitive (i.e., thinking, planning, scheduling, and organizing), this might explain higher rates of worries and ruminations in younger adults. Additionally, addressing the higher rate of negative repetitive thoughts in younger adults is the fact that Millennials tend to use the Internet more frequently to find medical information. More precisely, Beaudoin and Hong (2021) [50] mentioned that in 2017 and 2019, more than 90% of U.S. Millennials conducted a digital search for health or medical information, which is much more than Older Adults. Further, they also indicated that this proportion increased at the beginning of the pandemic to reach 99% of U.S. Millennials! At this point, it seems relevant to mention that an “infodemic” was ongoing in parallel with the health pandemic. According to the World Health Organization (2022) [61], an infodemic consists of “*too much information including false or misleading information in digital and physical environments during a disease outbreak*”. As Millennials are digital natives, it is likely that they were more prone to suffer from this infodemic than Older Adults. This observation elicits clinical perspectives in terms of prevention in the use of digital tools with younger adults.

Concerning results on intolerance of uncertainty, our study did not replicate previous findings. Indeed, intolerance of uncertainty did not vary across generations, while this psychological phenomenon has been identified as higher in the younger population by Glowacz and Schmits (2020) [16]. Even though no significant difference was found on this variable, it seems noteworthy to mention the difference between the “trait” of being intolerant to uncertainty and the “emotional state” of feeling distressed due to uncertainty. This nuance deserves to be thoroughly addressed in future studies. Indeed, if the repetitive exposure to uncertainty causes emotional distress, this might render individuals intolerant to uncertainty. From a clinical perspective, this could indicate the need for individuals to be able to handle their uncertainty without being overwhelmed by distress.

Despite sanitary restrictions that may have been more deleterious for younger adults, environmental satisfaction was similar in both generations. Nonetheless, it is noteworthy that Older Adults seemed more prone to engage themselves in novel activities. Indeed, we evidenced that a more important proportion of Older Adults started COVID-related helping activities such as doing grocery shopping for someone else or sewing facial masks. These activities can be considered as prosocial behaviors. Such behaviors have been proven to be linked to positive mood and life satisfaction [62]. In other words, in the context of the pandemic, adopting prosocial behaviors might have been protective against negative emotions that could be induced by the particular health circumstances. So, these helping behaviors were parallel to lower levels of negative effects. This might also be explained by the rationale underlying behavioral approaches to mood [25,26,27]. Environmental satisfaction—which is positively associated with good mood—can be reached through the contact with sources of pleasure and/or reinforcement [21]. It might be hypothesized that engagement in the above-mentioned activities for Older Adults is coherent with personal values and, therefore, consists of deep sources of pleasure and/or reinforcement, which in turn maintained their environmental satisfaction and prevented mood deterioration.

To sum up, our results confirmed phenomena that have already been evidenced within, but also outside the pandemic. The replication of previously obtained results is particularly important as our discipline is going through a “replication crisis” [63].

### 4.1. Clinical Implications

Our clinical recommendations will address issues relative to prevention, detection, assessment, and intervention in the area of mental health. Even though these recommendations will mainly be designed to help Millennials—as they reported higher levels of psychological distress—they remain relevant for Older Adults experiencing similar complaints and needs. The overreaching goal of offering clinical perspectives is to restrain—nay, avoid—another pandemic, of a psychological nature this time. To do so, we should offer evidence-based clinical practice following the pandemic. Our paper falls within this perspective.

We confirmed the experience of negative emotions in Millennials during the second wave of COVID-19, which was a less investigated period of the pandemic. Although uncomfortable, these emotions mobilize and incite people to change. Indeed, Bigot et al. (2021) [64] and Harper et al. (2020) [65] have shown that negative emotions encourage people to respect barrier gestures. Even though negative emotions could be useful to handle the dissemination of COVID-19, it remains necessary to alleviate them to prevent the development of more severe mental health issues. Indeed, long lasting negative emotions consist in one of the most important risk factors to mental illness [66]. Decreasing negative emotions could be achieved using usual clinical tools while they might need some adaptations to respect ongoing sanitary recommendations. Moreover, faced with the increase in the prevalence of psychological distress [3,4], it seems useful to invest in the development of prevention tools. Indeed, as suggested by Wagener, Stassart, and Etienne (2022) [8], delivering an intervention early enough could reduce mental health burden in Millennials. These first line interventions are usually provided through paper self-help brochures. However, as we address “digital natives”, it might be more appropriate to develop self-help websites or applications. The use of such digital tools has already been investigated for different psychological issues, outside any pandemic situation, and has proven efficient (e.g., [67,68]). Further, another interest in the use of digital tools resides in the fact that they might make Millennials more aware of their proneness to suffer from the “infodemic”. As the development of self-help websites or applications should be implemented in an evidence-based approach, the pieces of information they contain should, in turn, be reliable. Then, this might consist in a manner to teach Millennials to use accurate sources of information, in a suitable manner. Pursuing a similar aim, the World Health Organization has already developed different resources to help individuals to find accurate health information.

According to our results, Millennials should benefit from psychological support to handle their negative repetitive thoughts. This seems very important, as such thoughts predict the maintenance of anxious symptoms [53]. Coping with negative repetitive thoughts might be achieved by the recourse to more concrete ways of thinking [69]. Further, disengaging from the emotional response elicited by these ways of thinking might be eased by the practice of mindfulness [70]. In fact, meditation and mindfulness have been proven to be beneficial during crises periods [71]. Additionally, digital natives might appreciate smartphone applications which are already offered in mindfulness programs.

Based on the apparent protective effect of engaging in novel activities seen in Older Adults, Millennials might find some interest in the engagement in novel activities. This is in line with behavioral activation’s rationale aiming at (re)engaging oneself in pleasant and/or mandatory activities [72,73]. Even though behavioral activation principles are empirically grounded, it might be complicated to implement those principles in the context of a sanitary crisis. Indeed, behavioral activation principles usually imply the actual access to several activities, some of them happening outside homes and in presence of other individuals. Briefly, this highlights the need of remaining creative—while respecting recommendations—to reach our goals even though sanitary restrictions are applied. Our study showed that Older Adults engaged themselves in a particular kind of behaviors, namely prosocial behaviors. According to the framework around life values, Millennials should only start such activities if they are meaningful to them.

Overall, Older Adults seem to handle the pandemic situation more easily. Therefore, it could be useful for Millennials to benefit from an intergeneration’s communication on the former’s strategies to go through crises. Indeed, Older Adults have lived more life experiences than Millennials. In turn, it seems likely that they have a larger range of problem-solving strategies that they might be pleased to share with younger individuals.

### 4.2. Experimental Implications

Future studies should thoroughly investigate the use of digital media by Millennials to confirm our hypotheses according to which their use influences both negative emotions and negative repetitive thoughts. Future studies should also assess the impact of the COVID-19 pandemic on the autobiographical memory and, more specifically, on self-defining memories. These consist of memories concerning events or life-periods which occurred at least a year before the time of recall, in particular autobiographical memories, as they are integrated memories related to a meaning-making (i.e., a learned lesson about oneself, others, or the world) [74]. Then, they reflect central goals, values, and conflicts of one’s life. It might be enlightening to learn more about the differences in self-defining memories between generations as their views are shaped by their shared experiences. Further, based on Kwon, Eoh, and Park (2020) [75], mediation analyses assessing the impact of emotion regulation strategies on the relationship between emotion and behaviors might be relevant.

### 4.3. Limitations

Our results should be interpreted in the light of four limitations. First, psychological reactions were assessed with self-report measures, which might be prone to response bias, notably due to the conditions in which the participants fulfilled their questionnaires. In line with the previous, self-selection bias might also be active in this study, as recruitment occurred online. Second, the current study relies on data obtained in a cross-sectional design. Then, they only show a snapshot of COVID-19 psychological reactions at a particular time and space. Thirdly, our sample is mainly composed of women; our results might be quite different in a more balanced sample. Fourthly, we do not have information concerning the occurrence of a diagnosis of mental health prior to the pandemic, while they have been shown to influence one’s reactions to COVID-19 pandemic [76].

## 5. Conclusions

Overall, Millennials seem to experience more intense emotions, positive and negative, than Older Adults who were able to engage themselves in prosocial behaviors. Several explanations of these phenomena have been discussed. This paper also discussed clinical implications and experimental ones which aim at being generalizable at any time of crises. This consideration seems highly relevant, as we appear to live in a period of crises of different kinds.

## Figures and Tables

**Table 1 ijerph-19-05908-t001:** Sociodemographic characteristics.

		Millennials		Older Adults		Total Sample	
		*n*	%	*n*	%	*n*	%
Gender	Male	19	20.2	30	25.2	49	23.0
	Female	75	79.8	89	74.8	164	77.0
Marital status	Single	19	20.2	21	17.6	40	18.8
	Single, with children	4	4.3	15	12.6	19	8.9
	In a relationship	39	41.5	31	26.1	70	32.9
	In a relationship, with children	32	34.0	52	43.7	84	39.4
Working status	As usual	45	47.9	41	34.5	86	40.4
	Homeworking	42	44.7	45	37.8	87	40.8
	Technical lay-off	2	2.1	6	5.0	8	3.8
	Unemployment	1	1.1	1	0.8	2	0.9
	Student	2	2.1	1	0.8	3	1.4
	Other (i.e., retired, on sick leave)	2	2.1	25	21.0	27	12.7

**Table 2 ijerph-19-05908-t002:** Results of generations’ comparisons through independent *t*-tests.

			Millennials		Older Adults		*t*(211)	*p*	Cohen’s *d*
		Range	*M*	SD	*M*	SD			
Age		24–79	31.3	4.43	53.8	8.95	22.4	<0.001	3.09
Emotions	Joy	1–7	4.06	1.04	3.60	1.11	−3.10	0.002	−0.43
	Anger	1–7	3.10	1.24	2.80	1.29	−1.71	0.09	−0.24
	Fear	1–7	3.70	1.45	3.16	1.43	−2.73	0.007	−0.38
	Sadness	1–7	2.57	1.35	2.19	1.27	−2.15	0.03	−0.30
	Guilt	1–7	1.63	0.82	1.39	0.68	−2.29	0.02	−0.32
	Disgust	1–7	2.35	1.63	2.31	1.69	−0.16	0.87	−0.02
Emotions regulation strategies	Worry	1–5	3.02	1.05	2.68	1.00	−2.38	0.02	−0.33
	Dramatization	1–5	2.17	1.09	2.09	1.23	−0.48	0.63	−0.07
	Rumination	1–5	2.71	1.18	2.28	1.12	−2.75	0.006	−0.38
	Positive reappraisal	1–5	3.69	1.03	3.83	1.01	0.99	0.32	0.14
	Acceptance	1–5	3.73	1.09	3.85	1.08	0.77	0.44	0.11
Intolerance of uncertainty		1–5	3.33	0.80	3.16	0.85	−1.52	0.13	−0.21
Environmental satisfaction		1–5	3.51	1.03	3.69	1.02	1.33	0.19	0.18

**Table 3 ijerph-19-05908-t003:** Contingency table representing activities depending on generation’s belonging.

	Millennials		Older Adults		Total Sample	
	*n*	%	*n*	%	*n*	%
Artistic activities (e.g., painting and sculpting)	9	9.6	8	6.7	17	8.0
Sporting activities (e.g., running and biking)	14	14.9	10	8.4	24	11.3
Gardening	4	4.3	10	8.4	14	6.6
Cooking	13	13.8	11	9.2	24	11.3
Well-being activities (e.g., mindfulness and yoga)	3	3.2	13	10.9	16	7.5
COVID-related helping activities (e.g., grocery shopping for someone else and stitching facial masks for other people)	0	0	12	10.1	12	5.6
Not applicable	66	70.2	81	68.1	147	69.0

## Data Availability

Data are available upon a request sent to the corresponding author.

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
