# Peer review of "At the Peak of the Second Wave of COVID-19, Did Millennials Show Different Emotional Responses from Older Adults?"

_ijerph, 2022, doi:10.3390/ijerph19105908_

Round 1
Reviewer 1 Report
The authors deal with an issue widely addressed because of the pandemic, albeit from a different and interesting perspective: comparing millennials to older adults during the lockdown.
The work has some merits and has many limitations, as the authors themselves admit.
Major concerns
Why the authors based their sample size calculation on T-test for indip samples? How did they calculate alpha for multiple comparisons?
In the para 2.2.2., the authors state: "Then, we assessed the remaining life areas in which individuals were able to start novel activities (i.e., physical and psychological health issues, hobbies/recreation, volunteer work/charity/political activities, daily responsibilities). "
This choice would need better clarification. The absence of many of the participants' activities of daily living also had a negative effect on the
other behaviors examined by the authors. And with a different impact if we consider adults vs young people. This point should be clarified and discussed
Discussion
I would also leave more room for the fact that adults have greater ability to compensate for stressful events (Lazarus...) and resilience towards perceived risk. In this difference also lies an additional limitation of the study.
Author Response
Dear Reviewer,
You will find in the attached file, our responses to your comments.
We thank you for your comments which enable us to improve the quality of the manuscript.
Best regards.

Reviewer 2 Report
At the peak of the second wave of COVID-19, did Millennials actually go through the pandemics differently from Olders Adults?
Comments to Authors
The current study aimed to compare psychological reactions to the COVID-19 pandemic between Millennials – _adults born between 1981 and 1996 – _and Older Adults. The paper provides sufficient scientific contribution to the panorama of studies investigating the impact of COVID-19 pandemic on the general population's mental health, with a particular focus on the different generations.
The topic is interesting enough and qualitatively good to be published. However, some changes and explanations to improve the quality of the manuscript for its acceptance should be done.
Title and abstract section
The Authors would like to insert "title" terms such as "emotional responses". At the peak of the second wave of COVID-19, did Millennials show different emotional responses from Older Adults through the pandemics?
In the abstract section, the Authors should briefly describe the methodology used to recruit, assess, and analyze the considered variables (i.e., the instruments and sociodemographic, personal, emotional/psychological dimensions investigated by the survey).
Introduction section
In the 'introduction section,' the bibliography is appropriate and updated. However, it would be useful to mention some recent studies on the impact of the pandemic on the mental health of young and older adults (listed below). More specifically, some authors investigated emotional and cognitive responses in a sample of university students and teachers during the national COVID-19 lockdown. Home confinement compromised the possibility of fully experiencing university life, influencing academic study (i.e., uncertainties about cancellation, delays in activities, and digital platform use) with a significant impact on anxious symptoms, feeling of loneliness, sleep disorders, concentration difficulties, failure to achieve curricular objectives and uncertainty and worries about the future. Through a qualitative analysis of narrative diaries, the Authors identified, among other cognitive styles, including "Intolerance of uncertainty" (the most represented), the "all or nothing" as the strongest predictor of traumatic distress in a sample of young people. In addition, among others, the Authors found the changes in study context and habits and learning concentration impairment during Distance Education (DE) as the most substantial predictive factors of difficulties in academic performance. In another study, even the teachers with a mean age of 56,13 (SD=10,5) showed various levels of emotional distress and depressive symptoms, and difficulty concentrating, especially female teachers. 15.1% reported mild depressive symptomatology, 7.5% moderate depressive symptomatology, and 3.2% severe symptomatology. The appreciation of DE seemed related to the better emotional well-being of this teachers sample. The cited studies could be integrated into the manuscript because the authors conducted investigations during the first phase of the pandemic, using both quantitative and qualitative measures, an added value to their research contribution. Line 52. The sentence "disturbing emotions" would be more appropriate to be replaced with "emotional disorders" or "emotional distress". Lines 142-143 "The sentence "From a clinical perspective, the results of the current study might strengthen the need 142 of having at our disposal efficient clinical tools to avoid mental health worsening (e.g., 143 prevention, detection, assessment, intervention)" should be placed by Authors in the concluding section as a "Perspective" or "Conclusions" (already presents in the text) but not in the "Introduction section". The Authors should remove it.
Materials and Methods section
Authors should insert the "study design section" as the first subsection of the "Materials and Methods" relating to power analyses and, subsequently, "Participants and procedures section".
The survey was conducted online through a link, but it is unclear what type of form the Authors used (i.e., Google form, Microsoft form ….)
The tables are exhaustive for accurate data presentation.
Discussion of the Study
The section is very descriptive and redundant. The Authors should lighten the text for this purpose.
The clarity and structure of the manuscript are overall good enough, as the style of the language, the syntax, and the sentence construction. No editing of the English language is required.
References
https://pubmed.ncbi.nlm.nih.gov/34107926/
https://pubmed.ncbi.nlm.nih.gov/33384623/
https://pubmed.ncbi.nlm.nih.gov/34254057/
https://pubmed.ncbi.nlm.nih.gov/32229390/
https://pubmed.ncbi.nlm.nih.gov/32805704/
https://pubmed.ncbi.nlm.nih.gov/34708828/
Author Response
Dear Reviewer,
You will find in attached file our responses to your comments.
We thank you for your comments which help us to improve the quality of our manuscript.
Best regards.

Reviewer 3 Report
This is a very interesting article presenting the results of research conducted during the Covid-19 pandemic (At the peak of the second wave of COVID-19, did Millennials actually go through the pandemics differently from Olders Adults?). The design of the article corresponds to the texts that are based on empirical studies and contains all the elements of the structure of this type of study. The terminology is explained comprehensively based on the properly selected literature on the research subject. The research objective has been achieved and the presented results are cognitively interesting and prove good methodological skills of the authors.
Suggestions for authors:
- I propose to separate the purpose and hypotheses of the research.
- I suggest adding sample items in the measurement.
- It might have been interesting to include a mediation analysis that considered the mediating role of emotion regulation between emotion and behavior. We can hypothesize such results based on previous studies: 10.3389/fpsyg.2020.01114; 10.1111/1556-4029.14807
- Suggests a separate section: future research.
Author Response
Dear Reviewer 3,
The reply to your comments are in the attached file.
Thank you for your help which helps us improve the quality of the paper.
Best regards.

Reviewer 4 Report
Dear Authors,
this is interesting paper to read and review. In the following you will find some suggestions for you to revise your paper before its publication, by making implicit messages explicit.
Abstract
Please, include a mention about your research methods in the abstract.
Introduction
I see that your lengthy Introduction, dealing with quite many themes, as a multilayer compilation that can be clarified easily. As the paper concentrates on psychological level issues and looks the other aspects of human life through the lenses of the outcome for mental health, the broader contexts appear to be a mixture of triggers, causes or supportive mechanisms.
The clarification can take place just by reorganizing, preferably with numbered subtitles.
A good start would be to present the Covid-19 pandemic with the timelines of the waves, at its societal level. This would give a justification for your topic “At the peak of …” . It should be mentioned that the lockdowns and other preventive measures were launched to save lives and to protect health, not to cause mental health issues that evolved, could we say, as “unwanted side effects”. Please, remove the significant background issues from parenthesis (lines 29 – 30) to the text and explain them further. Also, in the very beginning of Discussion, in the lines 267 – 278, you present background issues and express the significance of your planned contribution in a way that I prefer to see in Introduction.
From the societal level the paper could proceed to everyday life, psychological levels, and experiences of the generations.
Introduction should end with clearly defined research tasks, using the approaches/variables derived from the literature presented.
Methods and Results
I appreciate the application of Power Analysis in determining sample size, since it contributes to an economic way to gather data. Methods -section is presented clearly and conveys consistency to Results -section.
Discussion
I suggest that your Discussion would include only those matters on how your methodological and theoretical approaches as well as variables chosen worked in the empirical research, compared to earlier research. With this I mean the first, unnumbered part of your text. Please, make sure to emphasizes any new and conflicting findings.
Please, think about presenting your perspectives, numbered as 4.1. and 4.2. as a new section titled for example as Implications.
All the best for your revisions
Reviewer
Author Response
Dear Reviewer,
You will find in attached file the reply to your comments.
We thank you for your comments which help us to improve the quality of the paper.
Best regards.

Round 2
Reviewer 2 Report
Thank you for your revision.